# Advances in Mass Spectrometry-Based Single Cell Analysis

**DOI:** 10.3390/biology12030395

**Published:** 2023-03-02

**Authors:** Siheun Lee, Hung M. Vu, Jung-Hyun Lee, Heejin Lim, Min-Sik Kim

**Affiliations:** 1School of Undergraduate Studies, Daegu Gyeongbuk Institute of Science and Technology (DGIST), Daegu 42988, Republic of Korea; 2Department of New Biology, Daegu Gyeongbuk Institute of Science and Technology (DGIST), Daegu 42988, Republic of Korea; 3Center for Scientific Instrumentation, Korea Basic Science Institute (KBSI), Cheongju 28119, Republic of Korea; 4New Biology Research Center, Daegu Gyeongbuk Institute of Science and Technology (DGIST), Daegu 42988, Republic of Korea; 5Center for Cell Fate Reprogramming and Control, Daegu Gyeongbuk Institute of Science and Technology (DGIST), Daegu 42988, Republic of Korea

**Keywords:** mass spectrometry, single-cell analysis, proteomics, metabolomics, mass spectrometry imaging

## Abstract

**Simple Summary:**

Cell-to-cell variation exists within a population of the same cell type due to stochastic gene and protein expression and environmental factors. Studying such cellular heterogeneity is the key to understanding the underlying mechanisms of fundamental biology and complex diseases, highly demanding developments in advanced technologies for molecular profiling at the single-cell level. The growth of single-cell RNA sequencing techniques has significantly contributed to this field. Moreover, the development of mass spectrometry (MS)-based methods for analyzing proteins and metabolites in a single cell is rapidly progressing. This review aims to present recent advances in MS-based single-cell analysis and their applications in biology and medicine.

**Abstract:**

Technological developments and improvements in single-cell isolation and analytical platforms allow for advanced molecular profiling at the single-cell level, which reveals cell-to-cell variation within the admixture cells in complex biological or clinical systems. This helps to understand the cellular heterogeneity of normal or diseased tissues and organs. However, most studies focused on the analysis of nucleic acids (e.g., DNA and RNA) and mass spectrometry (MS)-based analysis for proteins and metabolites of a single cell lagged until recently. Undoubtedly, MS-based single-cell analysis will provide a deeper insight into cellular mechanisms related to health and disease. This review summarizes recent advances in MS-based single-cell analysis methods and their applications in biology and medicine.

## 1. Introduction

The heterogeneity of cells in populations caused by cell-to-cell variation makes it necessary to analyze single cells and this will allow one to discover hidden mechanisms not seen in bulk samples (Figure 1) [1,2]. Recent studies utilizing both antibody-based methods and mass spectrometry-based methods successfully demonstrated the importance of single-cell analysis [3,4]. These single-cell analysis methods have been making rapid progress in terms of higher sensitivity as well as increased identification numbers and specificity. Single-cell analysis is essential for a deeper understanding of cancer, immunology, and other fields requiring precise information on cellular mechanisms. Diverse molecules in cells, such as RNA, proteins, and metabolites, can be analyzed at the single-cell level.

Among common approaches for single cell protein analysis are antibody-based methods. They are characterized using specific antibodies that bind to target proteins, which can then be identified through several techniques. Immunocytochemistry (ICC) is one such approach in which cultured cells or individual cells that have been isolated are tagged using an antibody of interest [5,6]. The antibody is linked to a reporter, usually a fluorophore or enzyme, which can then be detected in a microscope after fluorescence or color from an enzymatic reaction occurs. Another immunofluorescence method is immunohistochemistry (IHC), which differs from ICC in the fact that the cell staining is applied to intact tissue sections [7]. With ICC, most of the extracellular matrix and interstitial components are removed, leaving only isolated cells to be analyzed. For the conventional immunofluorescence analysis of single cells, ICC is generally used as it includes a cell isolation procedure. However, a comparison of the results between ICC and IHC may provide some insight into the differences between single cells and bulk tissue samples regarding the distribution of specific antigens. Another widely used method is fluorescence activated cell sorting (FACS). This method utilizes laser-induced fluorophores to count cells of interest based on antibody-antigen interaction. It has been reviewed elsewhere [8,9] and will not be discussed here. One critical limitation of antibody-based methods is that there must be antigen-specific binding between the target protein and antibody, which requires the rigorous testing of specificity [6].

Next-generation sequencing (NGS) has also proven to be a powerful tool in the analysis of single cells [10,11]. Whole genomes can be sequenced within a day and the increased sensitivity can lead to the detection of genetic alterations, such as somatic variants. In addition, RNA sequencing (RNA-Seq) can be used for the discovery of novel RNA variants and splice sites, as well as the quantification of mRNAs for gene expression analysis [12]. Single-cell RNA sequencing (scRNA-seq) allows for new biological discoveries, which otherwise would be unobtainable using traditional methods that analyze pooled bulk RNAs from tissues. These include the identification of rare cell types [13,14], gene regulatory networks inference [15,16,17], and cell type hierarchy reconstruction [18,19].

Mass spectrometry (MS) has been an essential tool for analyzing single cells [20,21]. Although it is the most powerful method for protein analysis, there have been challenges in its application to single cells. However, advances in simple, multiplexed, automated, and scaled-down sample preparation have opened doors for rapid analysis with high sensitivity. Along with single-cell proteomics, MS has also been used to identify and quantify metabolites and lipids at the single-cell level. Here, we will review recent progress and applications of MS-based single-cell analysis methods based on LC-MS/MS, with or without TMT labeling, and CyTOF for single-cell proteomics, followed by a discussion on ionization/desorption methods for single-cell mass spectrometry imaging (MSI), such as secondary ion mass spectrometry (SIMS), electrospray ionization-based MS (ESI-based MS), and matrix-assisted laser desorption/ionization (MALDI). Applications of single-cell proteomics and metabolomics will be highlighted.

## 2. Label-Free Single-Cell Proteomic Analysis

Liquid chromatography tandem mass spectrometry analysis (LC-MS/MS) is one of the most effective methods in proteome profiling at the single-cell level. The rapid evolution of the LC-MS system over recent years has enabled label-free single-cell proteomic (SCP) analysis capable of identifying and quantifying more than 1000 proteins [22,23,24]. The current platforms for label-free SCP analysis were mainly developed with either Orbitrap or trapped ion mobility spectrometry time of flight (timsTOF) mass spectrometers. Cong et al. recently introduced an ultra-sensitive workflow that combines nanoPOTS (nanodroplet processing in one pot for trace samples), ultra-low flow liquid chromatography, and high field asymmetric ion mobility spectrometry (FAIMS) with an Orbitrap Eclipse Tribrid mass spectrometer [23]. The workflow enhanced the depth of single-cell proteome with 1056 identified proteins on average. Importantly, the employment of FAIMS has tripled the number of identified proteins compared to their previous research, indicating the critical contribution of ion mobility technology to the gas-phase separation [25]. The benefit of FAIMS was further elevated in a new method named transferring identification based on FAIMS filtering (TIFF) [24]. Besides the mass-to-charge ratio value (*m*/*z*) and the retention time, TIFF utilized the FAIMS compensation voltage (CV) as a third-dimensional characteristic of precursor ions for the peptide identification of single cells based on the match between run (MBR) algorithm. The efficiency of TIFF was demonstrated with the average of proteome coverage increased to over 1200 proteins. Together with FAIMS-Orbitrap mass spectrometer system, timsTOF mass spectrometer, with its remarkable sensitivity, has been used in several label-free SCP analyses [22,26]. Newly introduced timsTOF for single-cell proteomic analysis showed ten-fold increased sensitivity with the optimal setting, enabling the identification of 843 proteins on average in a single Hela cell [22].

Data dependent acquisition (DDA) is the most common scan mode in LC-MS/MS analysis [27,28,29,30,31]. DDA methods have been initially used for single cell proteomics analysis [23,25,32]. However, the DDA strategy only selects a small percent of precursors ion for tandem mass analysis, leading to low data completeness in numerous samples such as single cells. Data-independent acquisition (DIA) method recently gained attraction in label-free proteomics analysis on account of minimal missing values across replicates [33,34]. A new microfluid chip named SciProChip was developed for SCP analysis in DIA mode by Gebreyesus and his colleagues [35]. Applying SciProChip to DIA analysis resulted in the identification of approximately 1500 proteins on average from single cells with less than 16% missing values. In the timsTOF SCP, diaPASEF (parallel accumulation-serial fragmentation combined with data-independent acquisition) scan mode was established to maximize the number of precursor ions for tandem mass spectrometry [32]. This method has increased the number of quantifiable proteins (up to 2083) per single Hela cell with high completeness [22].

Although the proteome coverage in label-free SCP analysis remains low, several important cellular biological processes were observed. Proteome profiling from single PC9 cells showed several proteins involved in NSCLC pathways, such as EGFR, TP53, NRAS and MAPK [35]. In another research, proteins related to cell cycles were readily quantifiable in SCP data with several differentially expressed proteins upon drug treatment [32]. These results demonstrated the capability of using label-free SCP analysis in biological and clinical applications in the future.

## 3. TMT-Assisted Single-Cell Proteomics

Tandem mass tags (TMT) are isobaric chemicals used for the accurate and multiplexed quantification of peptides and proteins using tandem MS analysis (Figure 2) [29,36,37,38,39]. All multiple reagents of a typical TMT set have the same nominal mass and an identical chemical structure composed of a mass reporter, a mass normalizer, and an amine-reactive group (Figure 2a). Each mass reporter of the reagents contains stable isotopes distinctly configured in the chemical structure, thus having different masses from one another. For TMT-based multiplexed proteomics, digested peptides from different samples are labeled with TMT reagents (10, 16, or 18-plex [40]) and analyzed simultaneously by a single run of LC-MS/MS. The mass reporters are cleaved from the TMT-labeled peptides after fragmentation and distinguished based on their distinct masses specific to the samples. Their ion intensities measured in MS2 spectrum and corresponding peptide sequencing enables relative quantification of the peptides. Multiplexed analysis using TMT alleviates the variability of separate measurements, and the enhanced intensity of precursor ions, which are accumulated from identical TMT-labeled peptides of all samples, improves the quantification of proteins [41]. The labeling method with TMT detected changes in the number of low-abundance proteins available for hypothesis testing, showing higher precision and fewer missing values compared to a label-free quantitation method [42]. It should be noted that the accurate interpretation of TMT-based quantitative proteomic data requires minimizing false positives, batch effects, and missing values [43,44].

TMT-based multiplexing technologies are especially useful in large-scale proteomics at the single-cell level demanding high sensitivity and high throughput [45]. However, to confidently identify and quantify thousands of proteins in an individual cell, a limited amount of the samples should be delivered to LC-MS/MS instruments to the fullest. In efforts to minimize sample loss and increase sensitivity, various sample preparation methods have been suggested for reproducible proteomic analysis of low quantities [46,47,48]. Sample loss could be minimized by using organic cosolvents as alternatives to detergents, which circumvents cleaning and tube transferring steps [46], or by minimizing sample handling on a simplified nanoproteomics platform [47].

For TMT-assisted single-cell proteomics, Slavov’s group developed single cell proteomics by mass spectrometry (SCoPE-MS), where a set of hundreds of cells as a carrier is assigned to one TMT channel for labeling and analyzed together with the TMT-labeled proteomes of single cells (Figure 2b,c) [49,50]. The carrier sample with an ample number of proteins increases the signal of low-input samples such as single cells. This combination of samples reduces missing values during chromatographic separation and improves quantification. Cells were mechanically lysed by sonication in glass microtubes to minimize protein losses instead of using chemicals that may cause significant losses. A thousand proteins were quantified in single cells using SCoPE-MS, demonstrating the ability of ScoPE-MS to identify distinct cell types and study the relationship between mRNA and protein levels in single cell [49]. Their updated version of ScoPE-MS, ScoPE2, optimizes automated sample preparation and MS data analysis to further improve quantification and throughput with lower cost and hands-on time [51,52]. In ScoPE2, the minimal proteomic sample preparation (mPOP) method was introduced to lyse cells, which utilizes a freeze-heat cycle to extract proteins efficiently in pure water without a cleaning step [53]. mPOP preparing samples in multiwell plates enables parallel processing with reduced lysis volumes, thereby increasing sample throughput and reducing cost. They also developed methods for MS data acquisition optimization and data interpretation for peptide identification enhancement, improving quantification and proteome coverage. With such advances, SCoPE2 successfully quantified over 3042 proteins in 1490 single monocytes and macrophages, and the proteomic data analysis showed a gradient of heterogeneous proteome state of macrophages [51]. The workflow of SCoPE2 for multiplexed single-cell proteomics is described in detail elsewhere, enabling the analysis of ~200 single cells per 24 h with standard commercial equipment [52].

A nanoPOTS approach has been reported and combined with TMT labeling to boost processing efficiency and throughput for single-cell samples [54,55,56]. nanoPOTS is a chip-based processing platform for preparing small cell populations and utilizes a robotic system that performs picoliter-liquid dispensing and cell isolations (Figure 2b). The total processing volumes of a single droplet reactor decreased to less than 200 nanoliters. The sample then goes through evaluation, extraction/reduction, alkylation, Lys C digestion, trypsin digestion, surfactant cleavage, peptide collection, and TMT labeling, all within the same nanodroplet. This sample processing method could be performed in much smaller droplets and inside wall-less glass reactor of 1mm diameter (a total surface area of 0.8 mm^2^), corresponding to a ~99.5% reduction compared to a typical 0.5 mL sample tube (~130 mm^2^). nanoLC measurements of cultured Hela cells on the nanoPOTS platform with the match between runs (MBR) algorithm of MaxQuant [57] identified 3092 proteins from as little as ten cells [54]. Additionally, nanoPOTS outperformed vial-based preparation in peptide identifications by a 25-fold increase for ~10 cells, confirming its suitability with ultrasmall samples. Given the results from previous research showing that thousands of cells were needed for a proteome coverage of over 3000, high sensitivity MS measurement with a sample size of 10 cells is a noteworthy capability of nanoPOTS. Compared to the previous research showing that a proteome coverage of over 3000 required thousands of cells, high-sensitive MS measurement with a sample size of 10 cells by nanoPOTS made a significant breakthrough [58,59].

It has been further developed into a nested nanoPOTS (N2) for isobaric-labeling-based scProteomics with high sensitivity, successfully demonstrating reduced reaction volume, an increase in quantified proteins, and an increased number of single cells analyzed [60]. N2 chip consists of cluster arrays of nanowells to digest and label cells with TMT. Sample processing on the N2 chip facilitated TMT pooling and retrieval by adding a microliter droplet on clustered samples in one TMT set. Reducing the nanowell diameters from 1.2 mm to 0.5 mm, compared to the nanoPOTS chips, decreased total processing volume by 85% and facilitated the digestion kinetics of trypsin, augmenting the sensitivity and reproducibility of proteomics and resulting in 230% improved protein/peptide sample recovery. Another related innovation is an integrated proteomics chip (iProChip) that provides all-in-one functionality from cell input to complete proteomic sample processing [35].

One limitation of nanoPOTS is that it involves a specialized platform. The nanowells had to be fabricated by a photolithography-based microfabrication technique. In the experiment, a nanoliter-scale liquid handling system was home-built as well. In addition, most commercial LC autosamplers are incapable of sampling nanoPOTS-generated samples. As a result, the LC-MS measurements require procedures involving depressurizing/repressurizing the LC system and disconnecting/reconnecting high-pressure fittings. This manual operation is highly labor-intensive and requires extensive expertise to avoid leaks and achieve reproducible sample loading.

Currently, many single cell studies are utilizing nanoliter dispensers to isolate and prepare single cells [52,61]. There has been an increasing demand for a high-sensitive single-cell analysis platform, and such a tendency will continue. To maximize data acquisition from a set number of unicellular samples, an efficient sample processing platform is necessary to minimize sample loss and boost sensitivity [62]. nanoPOTS can provide such a high-performance platform for single-cell proteomics analysis with a limited number of cells. In addition, the open-space architecture of the nanowells opens opportunities for the additional incorporation of LC-MS platforms and isolation technologies, such as fluorescence-activated cell sorting (FACS) [35] and laser capture microdissection (LCM) [63]. Innovations are being made to address the drawbacks of nanoPOTS. In one case, an approach that involved prepopulating the nanowells with DMSO and integrating nanoPOTS with LCM increased spatial resolution, resolving some of the procedural issues of microsampling [64]. Other relevant examples could be the N_2_ chip and iProchip implementing an autosampler. With additional developments to make it commercially available, nanoPOTS may become the platform of choice for single cell omics.

The nanodroplet platform can be efficiently applied to various single-cell studies although quantities of sample material are limited. Thus, the methods could be applied to proteomic studies of circulating tumor cells [65], stem cell development [66], high-level cellular heterogeneity [67], and biomarkers of disease [68].

## 4. CyTOF for Single-Cell Proteomics

Cytometry by time-of-flight (CyTOF) (Figure 3), or mass cytometry, is a variation of flow cytometry. Flow cytometry is a widely used immune profiling method in which a sample of cells is labeled with fluorescent markers and examined using a laser beam [69,70,71]. Each cell from the sample flows in a row through the laser excitation region, where multiple fluorescent markers of the single cells are measured simultaneously, and the fluorescence intensities represent a proxy of the expression level of the targeted antigens. Despite a high-throughput method used to analyze or quantify multiple cellular features at the single-cell level, there are limitations with flow cytometry. Fluorescence measurements often create spectral overlap and require compensation, limiting the number of parameters that can be measured together [72].

CyTOF resolves the overlapping spectral issue by replacing fluorescent probes with stable heavy-metal isotopes and combining flow cytometry with high-precision mass spectrometry to increase the number of cellular parameters to be quantified simultaneously. Such advances in mass cytometry provide insight into the cell subpopulations of complex cellular systems and their distinct functions [73]. The principles, workflow, and data processing of a novel CyTOF instrument are explained in detail elsewhere [73,74]. In CyTOF, single cells are isolated from a biological system and pooled in one tube. Target proteins are labeled with antibodies, each of which is conjugated with distinct heavy-metal isotopes from the lanthanide series not found in biological samples [75]. The labeled cells are nebulized into droplets and then directed into an inductively coupled plasma, which breaks the covalent bonds, producing heavy metal ions and small mass ions with masses below 75 Da. By filtering these small biological ions through a quadrupole, only the heavy metal ions will be introduced to a time-of-flight (TOF) mass spectrometry. With TOF analysis, the mass-to-charge ratio (*m/z*) of an ion is determined by the time it takes the ion to travel through and reach the end of the flight tube. Compared to fluorescent-based flow cytometry, the high mass resolution of CyTOF reduces spectral overlap between different metal ions, enabling high dimensional analysis of over 40 simultaneous cellular parameters for millions of single cells from a sample. Automated data analysis algorithms have been developed for CyTOF to aid in cell-subset clustering and phenotyping to provide biological insights [76,77,78]. Recently, CyTOF has been expanded into high-dimensional imaging techniques, imaging mass cytometry (IMC) (Figure 4a), with subcellular resolution by combining laser ablation to gain spatial information in tissues or cells stained with metal-tagged antibodies [79]. Basic principles, experimental workflows, and applications of IMC are explained in detail elsewhere [80,81]. One of the limitations of CyTOF as an antibody-based method is that it heavily relies on the availability and specificity of antibodies that bind to proteins of interest, which demands thorough validation of antibodies for a reliable analysis [82]. High-throughput analysis for single cells with high-dimensional information, including viability, cell morphologies, proteins, and even mRNA transcripts [83], makes CyTOF a powerful analytical technique to address biological questions in many applications, such as broad-scale immune profiling [84,85,86], T and NK cell subtyping [87,88,89], therapeutic responses [75,90], antiviral T cell response [91,92,93], biomarker discovery for diseases [94,95,96,97], and patient profiling involving COVID-19 [98,99]. CyTOF can also help to analyze the cellular heterogeneity related to clinical trials [100,101,102], autoimmune disorders [103,104,105], and cancers [106,107,108]. We will further discuss the current applications of CyTOF in the section on the application of single-cell proteomics.

## 5. SIMS for Single-Cell Mass Spectrometry Imaging

Secondary ion mass spectrometry (SIMS) (Figure 4b) is a surface analytical technique using energetic ion beams bombarded onto the surface of a specimen, leading to the ejection of atoms and molecules from the topmost layer, followed by ionization and measurement for surface composition, chemical imaging, and depth-profile analyses [109,110]. The ion beams are focused into up to tens of nanometers and rastered across sample surfaces, producing high-resolution chemical images without matrix applications. Moreover, 3D chemical mapping of a sample can be performed by alternatively repeating surface imaging and sputtering of the analyzed surface layer with minimum damages.

Since its advent, SIMS has been an indispensable analytical tool with surface specificity and high sensitivity in the fields of semiconductors, thin films, inorganic and organic materials [111,112,113]. However, high fragmentations due to energetic ion beams have long limited SIMS to analyzing mostly elements. Ion sources, such as liquid metal ion sources (LMIS) [114,115] and gas cluster ion beams (GCIB) [116,117,118], have been continuously developed in the past few decades to reduce the unwanted fragmentation of molecules, which makes SIMS capable of analyzing and imaging small molecules, e.g., lipids and metabolites in cells and tissues at the single-cell level [119,120,121]. The recent advance in SIMS imaging coupled with high-performance Orbitrap mass spectrometry (OrbiSIMS) now enables label-free molecular mapping of 3D biological samples with subcellular lateral resolution and high mass-resolving power [122,123]. Single-cell metabolomic profiling of macrophages by OrbiSIMS showed increases in phospholipid species and cholesterol with the accumulation of a drug [122]. Zhang et al. combined OrbiSIMS with cryogenic sample preparation methods to well-preserve water-containing samples such as biofilms, demonstrating increased sensitivity in analyzing polar molecules, e.g., amino acids in frozen-hydrated samples [123].

Despite the continued technical developments, SIMS has an intrinsic limitation in analyzing label-free proteins due to the unavoidable fragmentation of large molecules by energetic ion sources [124]. Yet, high-resolution multiplex protein imaging techniques are highly demanded to study the complex interactions of biomolecules in and between cells for an accurate understanding of molecular mechanisms in health and diseases. Among the most innovative methods to overcome the limitation of SIMS for protein imaging is multiplexed ion beam imaging (MIBI) which uses metal-tagged antibodies for staining clinical tissue sections and measures metal ions by an oxygen primary ion beam of high doses with high sensitivity [125,126]. MIBI can analyze over 40 labeled antibodies simultaneously over a five order of dynamic range and image them at subcellular resolution, which provides insight into how single cells are functionally organized and interrelated in a structured tumor microenvironment [126,127] or various pathological conditions [128]. While IMC has been readily adopted as a multiplexed imaging method due to easier accessibility, MIBI outperforms IMC in terms of sensitivity, spatial resolution, and analysis time. Further, a static SIMS-based multiplex protein imaging technique, which uses low ion doses to minimize ion beam damages on sample surfaces, has been introduced by labeling target proteins with metal oxide nanoparticles (MONPs)-conjugated antibodies for the simultaneous imaging of labeled proteins and label-free lipids in a hippocampal tissue [129]. Recently, Kotowska et al. showed that unlabeled proteins could be analyzed and imaged by OrbiSIMS, where a protein is fragmented into peptides by GCIB as a primary ion beam and identified via de novo peptide sequencing [130].

Since the use of accelerated ions in SIMS analysis requires an ultra-high vacuum environment, biological samples should be properly prepared by chemical fixation, freeze-drying, and cryofixation [131,132,133]. However, an artificial alteration of chemical composition, distribution, or structure could occur during preparation procedures, which demands developing novel methods. Lee et al. developed a facile sample preparation method using single-layer graphene and air-plasma treatment for improved TOF-SIMS molecular imaging of dried single cells with preserved morphology [134,135]. High-resolution TOF-SIMS lipid imaging of neuronal cells prepared by this method has been well-adopted for neuroscience studies [136,137,138]. Recently, it was reported that SIMS imaging of live cells using single-layer graphene can be performed even in an ultra-high vacuum environment for visualizing the intrinsic distribution of lipids in the intact cell membranes without significant distortion [139].

Advances in instruments, biochemistry, nanomaterials, and sample preparations altogether are needed to improve further the performance of high-resolution SIMS molecular imaging techniques for investigating the spatial distribution of biomolecules (proteins, lipids, and metabolites) and their complex inter-and intracellular interactions in the native state.

## 6. ESI-Based MS for Single-Cell Metabolomic Analysis

Electrospray ionization (ESI) is a soft ionization technique using an electrospray to produce gas phase ions from thermally labile biomolecules in solution for ambient mass spectrometry [140,141,142]. ESI applies a high voltage at the electrospray capillary, from the opening tip of which the Taylor cone of sample solution forms and emits an aerosol of highly charged droplets with the same polarity as the capillary voltage. As solvent evaporation continuously reduces the charged droplets in size, charge density on the surface increases, eventually transferring the charged nanodroplets into gas phase ions of analytes with multiple charges before entering a mass spectrometer. Basic principles, instrumentation, and applications of ESI-MS are described in greater detail elsewhere [143,144]. ESI ‘soft’ ionizes intact analytes without fragmentation, and multiple charging of resulting ions puts their *m/z* values within the mass ranges of mass analyzers, making ESI-based MS useful for the detection and structural analysis of large biomolecules, such as proteins and nucleic acids. In addition, ambient ionization of ESI-based MS can preserve the molecular contents nearly in the native state. With these advantageous features and the development of various single-cell sampling methods, ESI-based MS methods have been utilized and advanced for single-cell metabolomic analysis over the last decade [145], such as nano-ESI MS [146], probe ESI (PESI) MS [147], capillary electrophoresis ESI (CE-ESI) MS [148], desorption ESI (DESI) MS [149,150,151,152], nanospray DESI (nano-DESI) [153] MS, and LAESI MS [154,155]. Among them, DESI and LAESI can perform ambient desorption ionization for label-free MSI analysis of biological samples.

## 7. DESI for Single-Cell Metabolomic Mass Spectrometry Imaging

DESI (Figure 4c) pneumatically generates charged droplets and ions of solvent through ESI and shoots them to the surface of target analytes at a high velocity to desorb and ionize molecules of the analytes in the gas phase [156]. DESI MS has extensively benefited many applications of bioanalysis [157,158] and forensics [159,160], with a wide range of specimens to be analyzed, including plants [161,162], biofilms [163], zebrafish [164], human fluids [160], cells [165], tissues [166,167,168], food [169], etc. By moving a sample stage, DESI MSI is also possible, which allows one to obtain the spatiotemporal information on molecules directly from intact biological samples at ambient conditions; target molecules can be small ones, e.g., metabolites and lipids in singly charged form [170], or larger ones, e.g., proteins and peptides with multiple charging [171]. Due to the relatively poorer spatial resolution of DESI MSI (>50 μm) compared to the other MSI techniques, DESI MSI has been primarily used for molecular imaging of biological tissues [166,167,168] or metabolic profiling of entire individual oocytes and embryos [149,150,151,152]. The workflow for DESI-MS lipid profiling of individual oocytes and embryos, including classes of lipids obtained and data interpretation for a biological understanding of oocyte and embryo lipid metabolism in early development, is explained in detail elsewhere [152].

## 8. Nano-DESI for Single-Cell Metabolomic Mass Spectrometry Imaging

About a decade ago, Laskin et al. first developed nanospray DESI (nano-DESI) (Figure 4e) that utilizes a micro-liquid junction formed between two capillaries (a solvent delivery capillary and a nanospray capillary) and the sample surface where analytes are extracted and transported via the nanospray capillary to a mass spectrometer, improving sensitivity and spatial resolution [172]. Additionally, nano-DESI enables online quantification by adding internal standards to the extraction solvent [173,174,175]. Nguyen et al. coupled a nano-DESI MS system to shear force microscopy in order to precisely control the distance between the probe and the sample surface, enabling constant-distance imaging of biological samples with complex topography [176]. This advanced method has proved capable of quantitative and high spatial resolution (sub-10 μm) imaging of lipids and metabolites in lung, brain, and uterine tissue sections [177,178]. Bergman and Lanekoff used nano-DESI MS for the molecular profiling of lipids and metabolites in single cheek cells (60–90 μm in diameter), detecting and identifying 48 species, including phosphatidylcholine (PC), phosphatidylethanolamine (PE), sphingomyelin (SM), plasmalogen, amino acids, creatine, spermidine, and carnitines, via MS/MS [153]. They also quantified the total amount of PC in one cheek cell by adding a PC internal standard to the nano-DESI solvent. Although the enhanced sensitivity and spatial resolution of nano-DESI have enabled single-cell analysis, it still needs further improvement for molecular profiling or imaging at the subcellular level.

## 9. LAESI for Single-Cell Metabolomic Mass Spectrometry Imaging

LAESI (Figure 4e) is an ambient desorption/ionization method suited for water-containing samples, which not only has better spatial resolution than DESI but retains many advantageous features of DESI, such as minimal sample preparation and matrix-free ambient MSI analysis. Nemes and Vertes developed LAESI which utilizes a focused mid-infrared laser with a wavelength of 2.94 μm that excites water molecules to ablate analytes from the sample surface into the gas phase. The gas plume subsequently interacts with charged droplets generated from ESI, thereby ionizing the ablated analytes [179]. They showed that LAESI enables the identification and quantification of biomolecules, such as proteins (up to 66 kDa), lipids, and metabolites, and in vivo MSI of metabolites in a seedling. They further improved LAESI to be capable of 3D MSI of metabolites in biological tissues with a depth resolution of sub-40 μm [180,181]. Experimental procedures for ambient molecular MSI using LAESI are described in detail elsewhere [182]. LAESI MS has been widely applied to the direct analysis of intact samples, such as plants [183,184,185,186], rodent brains [187], body fluids [179], food [188,189], fishes [190], drugs [191], biofilms [192,193], etc.

IR laser-based methods like LAESI have more difficulty analyzing single cells than UV laser-based ones like MALDI as laser beam size increases with wavelength. Nonetheless, single-cell analysis has been possible using LAESI MS in combination with other techniques to improve sensitivity and spatial resolution up to the single-cell (10–100 μm) or subcellular level. Shrestha et al. demonstrated that LAESI MS enables the in-situ analysis of metabolites in plant tissues and animal eggs at the single-cell level (up to 30 μm) by delivering laser pulses through the etched tip of GeO_2_-based glass fiber, adjacent to the sample surface, to ablate single cells [194]. Using this optical fiber-based LAESI (f-LAESI), they identified a few tens of metabolites in single cells via MS/MS; the comparisons of single-cell metabolism showed that cell-to-cell variation existed within the same tissue depending on pigmentation or age and between cells of the same type in different species [194]. In situ cell-by-cell molecular imaging of metabolites in plant tissues using f-LAESI was also demonstrated [195]. Stolee et al. combined f-LAESI MS with microdissection to peel off the plant cell walls for the laser to selectively access the cytoplasm and the organelles, enabling subcellular analysis of metabolites in single cells [196]. This study showed that the metabolite compositions significantly differ between subdomains, such as the nucleus and cytoplasm, within a single cell. Stopka et al. integrated f-LAESI MS with fluorescence and brightfield microscopy to selectively sample single cells of a specific type in plant tissues, showing the potential of single-cell f-LAESI MS to study cellular heterogeneity and metabolic noise for hundreds of cells [154]. Recently, Samarah et al. employed f-LAESI MS combined with 21T Fourier transform ion cyclotron resonance mass spectrometry (21T FTICR MS) with ultrahigh mass resolution and mass accuracy to study isotopic fine structures (IFSs) of single-cell metabolites, increasing confidence in their identification [197]. Additionally, the Vertes group developed LAESI in transmission geometry (tg-LAESI) to analyze small adherent cell populations with spot sizes of 10–20 μm [198]. Taylor et al. developed a dual imaging system of optical microscopy and LAESI-MS to obtain structural and chemical information from single cells. They integrated the system with ion mobility separation (IMS) to increase the accuracy of molecular annotations through accurate collision cross-section measurements [199].

As stated, LAESI-MS has been continuously advancing for in situ molecular (mainly metabolites) profiling or imaging of single cells. Yet, the reduced spot sizes for high spatial resolution mean a limited volume of ablated samples, demanding more sensitive and efficient ways of sampling and ionization [196]. Although ambient ionization methods enable the in-situ analysis of biological samples in the native state, the surrounding environment could give sources of background, which makes molecular annotations challenging, leading to low molecular coverage. As briefly described above, some presented LAESI combined with 21T FTICR MS [197] or IMS [199,200] to tackle this issue. Currently, single-cell LAESI MS can analyze only hundreds of single cells, which is significantly low compared to the number of single cells analyzed in single-cell transcriptomics (100,000 cells) and single-cell proteomics (thousands of cells).

## 10. MALDI for Single-Cell Mass Spectrometry Imaging

Matrix-assisted laser desorption/ionization (MALDI) (Figure 4f) is a soft ionization technique that involves a laser beam and an energy-absorbing matrix to create intact ions of large molecules from target samples for analysis of biomolecules [201]. In MALDI MS, samples are generally prepared on a metal plate. A matrix, an organic compound or mixture, is uniformly applied onto the surface of samples and dried to form crystals that absorb laser energy to ablate and desorb the analytes from the sample, generating charged molecular ions with minimized fragmentation and decomposition. A high voltage is applied to the metal plate so that the charged ions can be accelerated and fly into a mass analyzer, e.g., TOF, Fourier transform (FT) ion cyclotron resonance or Orbitrap, to measure mass-to-charge ratios, producing a mass spectrum of the analytes. More details on the principles and workflows of MALDI MS methodology can also be found elsewhere [202,203,204]. Matrices are pivotal to successful MALDI MS experiments, and the advances in various novel MALDI matrices have been comprehensively reviewed in the literature [205,206]. MALDI MS is especially useful to analyze a mixture of analytes such as protein digests and polymers because singly charged ions are major forms produced during the process, which reduces the mass spectral complexity [207,208]. It also covers a wide range of sample types, including reaction mixtures [209], tissues [210,211] and biofluids [212,213,214,215]. In addition, there is no need to tag or pre-select the known targets of interest, unlike immunolabeling methods.

MALDI MS has been advanced in sensitivity, lateral resolution, speed, and cost/time efficiency, becoming a promising tool for single-cell analysis in health and diseases [216]. By minimalizing the fragmentations of analytes, MALDI MS can measure and image intact macromolecules such as lipids and peptides to create unique cellular profiles. Single-cell MALDI MS can provide valuable data regarding cell heterogeneity [217,218,219], drug metabolism [220], and patient diagnosis [221,222]. In addition, a combination of MALDI MS and immunocytochemistry may produce a complete characterization of polypeptide expression. These data, coupled with single-cell analysis of post-translational modifications [223,224], will provide invaluable insight into diseases and cancer. MALDI is also one of the widely used scanning techniques, including DESI, nano-DESILAESI, and SIMS, with soft ionization capabilities for the mass spectrometry imaging (MSI) of biomolecules, such as lipids, peptides, proteins, and other molecular contents in single cells and tissues [225]. We will discuss the current applications of MALDI for single-cell analysis in the following application sections.

The main limitation of MALDI is that it is challenging to analyze low molecular weight compounds due to the matrix-induced interference in the low mass range [226]. Matrix-free laser desorption/ionization (LDI) [227] can help detect small molecules in biological samples that retain laser-absorbing molecules inside, such as plants [228,229] and algae [230,231]. Single-cell analysis using laser-based MS is explained in greater detail elsewhere [21].

## 11. Application of Single-Cell Proteomic Analysis

Currently, single cell proteomic analysis, especially by the label-free method, can characterize several subpopulations of cancer cells based on their specific cellular processes. Gebreyesus et al. profiled specific pathways for non-small-cell-lung cancer (NSCLC) and lymphocytic leukemia cells with high coverage of related proteins [35]. Another research characterized four distinct subpopulations from Hela cells upon thymidine and nocodazole treatment, indicating heterogeneity in response to drugs for cancer [22]. Large-scale single proteomic analysis of macrophages observed several altered pathways under lipopolysaccharide treatment, demonstrating the impacts of SCP analysis in biological and clinical applications [24].

Immune cells also exhibit heterogeneity at the single-cell level. Understanding the cell–cell variation of immune cells is important because it provides insight into the many pathways the immune system uses to respond to pathogens [232]. Single-cell analysis revealed heterogeneity within macrophages [51], regulatory T cells [233], and monocyte cells [234]. Specht et al. developed SCoPE2 as an efficient method for TMT-assisted single-cell proteomics, revealing the emergence of macrophage heterogeneity as homogeneous monocytes differentiated into macrophage-like cells in the absence of polarizing cytokines [51]. The analysis of 3042 quantified proteins from 1490 monocytes and macrophages and comparing differential genes revealed a continuous gradient of proteome states for macrophages. By pairing the SCoPE2 data with 10× Genomics data, they uncovered regulatory interactions between the tumor suppressor p53, its transcript, and the transcripts of genes regulated by p53.

CyTOF is a practical tool for single-cell proteomics. The use of antibodies to simultaneously detect protein targets of interest enables the accurate, multidimensional identification of cell subsets. This analytical power of CyTOF has led to many biological applications in diseases and medicine. Tracey et al. highlighted notable findings of CyTOF for single-cell proteomics studies of mice for cancer, neurodegenerative diseases, inflammatory diseases, and infection [235]. These results identified changes in gene expression, signaling, and cell phenotypes during disease progression. Those findings on cell-to-cell variations can provide foundations to study disease mechanisms, potential biomarkers, and therapeutic targets. Mrdjen et al. used CyTOF analysis with a mouse model to examine aging as a risk factor in Alzheimer’s disease (AD) [236]. In repeated experiments, an identical phenotypic signature in a subset of microglia from AD mice was identified, thus providing a novel biomarker for AD progression. CyTOF has immense potential for therapeutic use as well. In diseases, it is crucial to understand how the balance of cell subsets between groups of patients or within one patient changes over time, how these subsets interact, and which are dysfunctional [237]. CyTOF technology can be very effective for disease research, as simultaneous measurement of cell surface markers and phosphorylation in catalysts of biochemical responses is possible, facilitating the identification of rare cell subsets and the observations of the off-target effects of drugs. CyTOF can also help improve treatment effectiveness. Anchang et al. developed a computational framework, drug nested effects models (DRUG-NEM), to analyze single-drug perturbation data with CyTOF, which can individualize drug combinations [238]. Single-cell data of tumor cell lines and primary leukemia samples, before and after drug treatment, were collected and used to quantify changes in the expression of intracellular markers across a panel of drugs. Then, nested effects modeling [239,240] was used to optimize drug combinations by choosing the minimum number of drugs that produce the maximal desired intracellular effects. DRUG-NEM could identify and select TRAIL and MEK inhibitors in HeLa cells and PI3k/mTOR and ABL/Src Inhibitors in acute lymphocytic leukemia cells as predominant optimal combinations [238]. Similar reports showed that profiling with CyTOF provides crucial information to discover novel drug combinations, maximizing treatment efficacy [241,242,243,244].

## 12. Application of Single-Cell Metabolomic Analysis

MS-based metabolomics is an emerging technology that can characterize the alteration of intermediate or final products of metabolism processes at the systems level. For the last decades, metabolomics has significantly contributed to decipher biological and clinical questions at the molecular level. Various platforms of MS-based technology have been applied to metabolomics studies, including SIMS, MALDI-TOF, LAESI-MS, and LC-MS [245,246,247,248]. Recent advances in single-cell metabolomics (SCM) have accelerated the understanding of cell population heterogeneity. Nevertheless, only a limited number of metabolites are frequently identified in SCM. In addition, their complex structural and physicochemical properties remain challenging to identification and quantification. Despite several challenges, single-cell metabolomic analysis has been performed successfully in different mass spectrometry platforms [249].

Mass spectrometry imaging (MSI) has been one of the most successfully applied methods in single-cell metabolomic analysis thus far [250]. The significant enhancement in spatial resolution, specificity, and sensitivity of MSI in the last few years has enabled the characterization of several metabolites from a single cell. Moreover, SIMS is the highest spatial resolution method in MSI. Although the spatial resolution can be reduced to sub-micrometers, several ion beam compositions cause molecular fragmentation [251,252,253]. The development of a softer desorption method called gas cluster ion beam (GCIB)-SIMS has significantly reduced the fragmentation of molecules [254]. Further optimization of CO_2_ GCIB brought the spatial resolution to 1 µm, which enhanced subcellular imaging with a distribution of several intact metabolites, such as phosphatidylethanolamine, cardiolipin, and purines [255,256]. Another widely used method in MSI-based SCM analysis is the MALDI MSI system. Several groups have developed different methods based on MALDI MSI to analyze the metabolome with cellular and subcellular resolutions [257,258]. However, most of available instruments have a pixel size in the range of 5–20 µm, hindering metabolome profiling at the single-cell level [259]. To improve the spatial resolution of MALDI MSI, a transmission geometry vacuum ion source was developed to minimize the laser spot size to less than 1 µm [260]. Smaller pixel size is obviously correlated with ion abundance reduction, which greatly affects the efficiency of metabolite detection from a single cell. Soltwisch et al. developed a post-ionization strategy named “MALDI-2” to improve the efficiency of metabolite detection from a small spot size [261]. Based on this strategy, Niehaus et al. further created a new system called t-MALDI-2, which not only reduced spatial resolution to 600 nm but also maintained the high sensitivity [259]. Together with high spatial resolution, MALDI-MSI has also seen substantial improvement in throughput. SpaceM method offered the ability in detection of over 100 metabolites from over 1000 cells per hour [262]. The application of MALDI-MSI to single-cell metabolomics is increasing these days in various research fields. Maria et al. applied this technique to analyze the lipid distribution in newly fertilized individual zebrafish embryos [263]. MALDI-MSI enabled the monitoring of several lipids of multiple myeloma (MM) at the single-cell level, identifying some specific cell types in MM progression [264].

## 13. Conclusions

There have been many innovations in techniques and methodologies to enhance sensitivity and resolution for the mass spectrometry analysis of single cells. The importance of studying cellular heterogeneity has led to such advances in single-cell analysis methods. The concept of cellular heterogeneity is widely accepted, describing that even within a population of a given cell type, phenotypical differences affecting cellular functionality and cell fate decisions are present [265]. Cell-to-cell variation is affected by several endogenous and exogenous factors, such as stochastic gene expression and microenvironmental changes [266]. Studies reveal that stochastic fluctuations or ‘noise’ in genetic circuits [267], along with cellular stress [268,269,270], trigger changes in gene expression, ultimately leading to cell-to-cell variation. As proteins are molecular executors of cellular functions, analyzing proteins at the single-cell level is fundamental in characterizing cell-to-cell variation and classifying cellular subpopulations [271]. MS is a powerful tool that can map whole proteomes. However, the current performances of MS for single-cell proteomics are insufficient in analyzing significantly limited amounts of a small-sized single cell due to the low instrument sensitivity and poor sample preparation. Continued developments in sample preparation, instruments, and data analysis may have enabled single-cell proteomic analysis with improved proteome coverage and high throughput.

Cellular heterogeneity has important biological and medical implications for diseases [272]. Genome-wide association studies (GWAS) suggest that genetic variants impacting various pathways and cell types within the same disease are responsible for drug ineffectiveness [273]. This issue can be addressed by characterizing different cell types and pathways at the single-cell level [272]. Cases of cellular heterogeneity and related therapeutic applications have been studied in cancer, autoimmune diseases, and other disorders. Intra-tumor heterogeneity, specified by differences in molecular signatures, has been observed in almost every type of cancer [274,275]. Variations in genetic signature, gene expression, and post-translational modifications at the single-cell level determine differences in cancer cell plasticity and growth [276,277]. Heterogeneity presents a challenge for cancer treatment. If a drug targets only a subpopulation of the tumor, relapse can occur due to tumor adaptation [274]. For example, epidermal growth factor receptor (EGFR) is a therapeutic target for non-small cell lung cancer (NSCLC). However, small subpopulations with resistance due to variation, such as EGFR mutations, are likely to progress [278,279,280]. Similar cases of therapeutic failure have been reported for thyroid cancer [281], renal carcinoma [282], and mammary tumors [283]. To overcome drug resistance driven by cellular heterogeneity, researchers must take into account a comprehensive characterization of single cells. Single-cell proteomics opens possibilities for therapeutic applications as well. For example, genetic heterogeneity in immune cells can cause differences in pathophysiology, clinical presentation, and response to therapies for rheumatoid arthritis (RA) [234,284]. Studies have shown that changes in macrophage prevalence can predict the effect of RA treatment. By utilizing single-cell proteomics data that captures changes in macrophage polarization, researchers can distinguish the biological functions of macrophages in different states and their characteristics in the pathophysiology of diseases.

To fully understand the underlying mechanisms of fundamental biology and complex diseases, reliable methods for obtaining single-cell multi-omic information (genome, transcriptome, proteome, and metabolome) are highly demanded. Capturing this information has been a considerable technological challenge for researchers. In the past decade, scRNA-seq technologies have rapidly advanced to profile individual cells and identify rare cell types based on gene regulatory networks for cellular heterogeneity study. Since mRNA transcript levels do not always correspond to protein levels due to several processes in a cell, such as transcription, mRNA degradation, translation, and protein degradation, it is necessary to profile protein expression at the single-cell level. MS has become a staple technique in proteomics and metabolomics, capable of effectively detecting, identifying, and quantifying proteomes and metabolomes. Recent advances in sensitivity, multiplexing, high throughput, data processing, and sample preparations for MS or MSI have facilitated a breakthrough in analyzing single-cell proteins and metabolites.

The methods for MS-based single-cell analysis covered in this review are effective techniques in collecting and interpreting multi-dimensional molecular information in hundreds to thousands of individual cells. The characteristics of those methods are summarized in Table 1. With the developments of robust workflows, including efficient sample preparation, combinations of advanced techniques, and automated data analysis, multiplexing technologies with or without TMT labeling have improved sensitivity, proteome coverage, and throughput, proving to be powerful tools for large-scale proteomics in many single cells. Antibody-based methods are well-combined with MS techniques, e.g., CyTOF and SIMS for the multifaceted analysis of single cells. High-dimensional imaging techniques with a subcellular spatial resolution, e.g., SIMS and MALDI, provide additional spatial information, which gives insight into organized structures of cell subpopulations in a tissue and the physiological implications thereof. Ambient desorption/ionization methods, e.g., ESI-based MS (DESI, nano-DESI, and LAESI) enable the direct metabolomic analysis of biological samples at the single-cell level preserved at ambient conditions (temperature, pressure, humidity, etc.) with minimal sample preparation. One of the future tasks would be to advance in the sample processing and analysis phase to improve sample throughput, sensitivity, accuracy, and quantification. The main goal of single-cell sampling is to quickly isolate as many cells of interest from biological samples as possible and efficiently send them to the analytical instruments with or without labeling. A successful procedure should guarantee a sufficient concentration of cells with high viability, minimal cell debris and aggregates, and preserved cell surface antigens. A possible approach to gain multi-omic data is integrating several single-cell analysis techniques for the same cells in a compatible and complementary fashion. Novel multi-modal technologies, advances in efficient sample preparation to minimize damage to the sample, and optimal workflow for robust multi-omic data acquisition will be key to the precise analysis of single cells.

## Figures and Tables

**Figure 1 biology-12-00395-f001:**
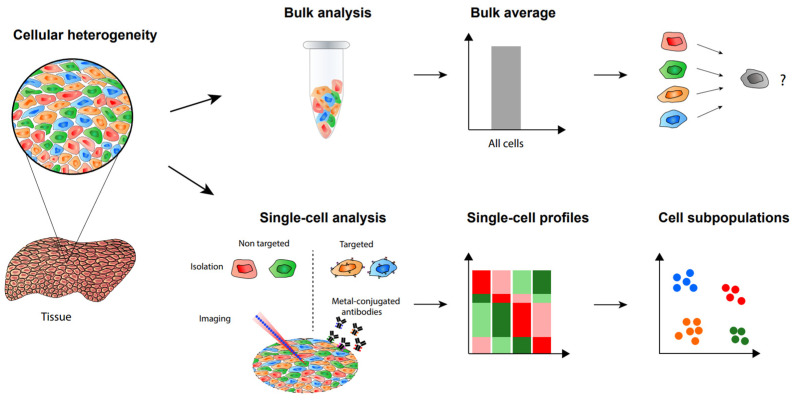
Single-cell analysis and imaging reveal cellular heterogeneity not seen by bulk analysis methods.

**Figure 2 biology-12-00395-f002:**
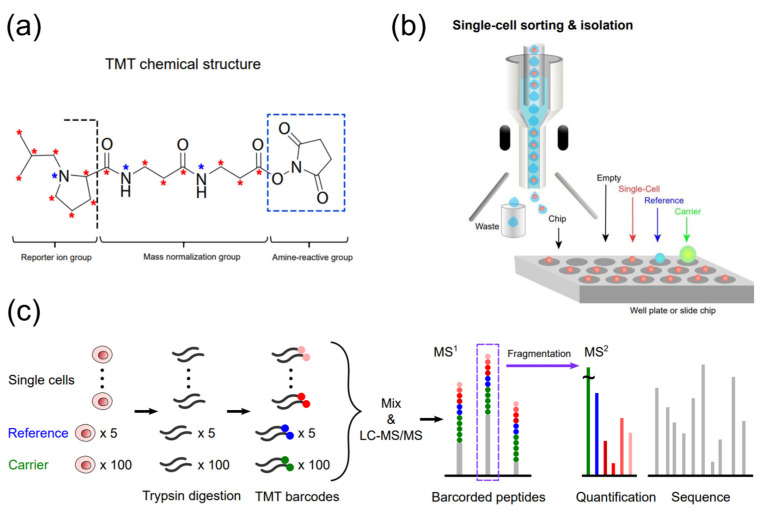
TMT-assisted single-cell proteomics. (**a**) The chemical structure of TMT (e.g., 18-plex), (**b**) Single-cell sorting and isolation onto a well plate or slide chip by FACS or a robotic system. (**c**) A typical workflow of single-cell proteomic analysis using TMT.

**Figure 3 biology-12-00395-f003:**
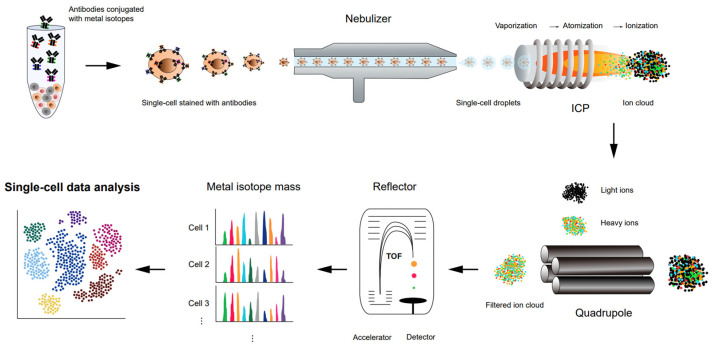
Schematic representation of a workflow of high-throughput single-cell proteomics analysis using CyTOF.

**Figure 4 biology-12-00395-f004:**
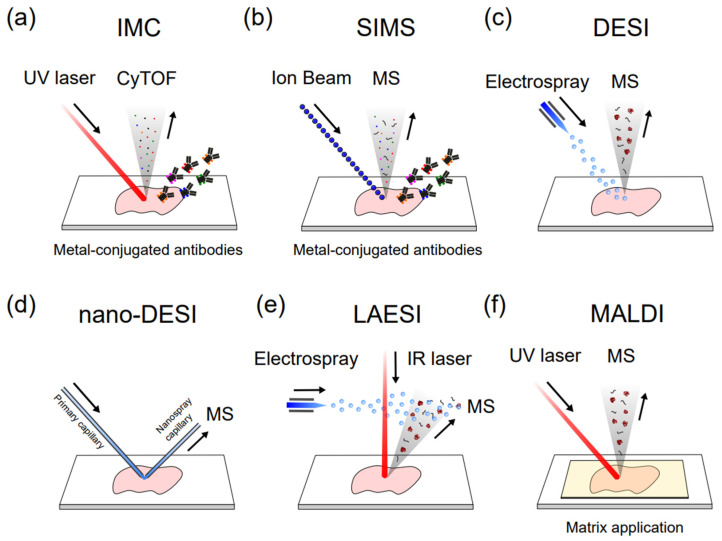
Different single-cell MSI methods. (**a**) Imaging mass cytometry (IMC); (**b**) Secondary ion mass spectrometry (SIMS); (**c**) Desorption electrospray ionization (DESI); (**d**) nanospray-DESI (nano-DESI); (**e**) Laser ablation electrospray ionization (LAESI); (**f**) Matrix-assisted laser desorption/ionization.

**Table 1 biology-12-00395-t001:** Comparative overview of different MS-based single-cell analysis methods. NA: not available.

	Target /Non-Target	Spot Size	Molecular Coverage		Single-Cell Throughput	Ambient /Vacuum	Limitations
			Metabolites	Proteins			
MS-based scProteomics							
Label-free	non-target	NA	NA	>1000	very low	ambient	low data completeness in DDA mode
TMT labeling	non-target	NA	NA	>1000	low	ambient	inaccurate quantification
CyTOF	target	NA	NA	~100 (metal-tagged)	very high	vacuum	availability of antibodies biased data acquisition
Single-cell MSI							
SIMS	non-target (TOF-SIMS) target (MIBI)	50 nm–50 μm	tens	MIBI: ~100 (metal-tagged)	high	vacuum	TOF-SIMS: low identification/quantification MIBI: Availability of antibodies
DESI	non-target	>50 μm	tens	limited	very low	ambient	low spatial resolution
nanoDESI	non-target	>10 μm	hundreds	limited	medium	ambient	delicate fabrication of probes
LAESI	non-target	>30 μm	tens	limited	medium	ambient	environment-induced noises
MALDI	non-target	1–25 μm	hundreds	tens	medium	vacuum	low detection of small moleculeslow quantification accuracy

## Data Availability

Not applicable.

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
