# Peer review of "Advances in Mass Spectrometry-Based Single Cell Analysis"

_biology, 2023, doi:10.3390/biology12030395_

Round 1

Reviewer 1 Report

The authors of the manuscript reviewed the recent progress in mass spectrometry-based single-cell analysis. They comprehensively discussed the pros and cons of the four different technologies (TMT, CyTOF, SIMS and MOLDI) used in single-cell proteomics and metabolomics analysis. This review comprehensively summarized the recent breakthrough in MS-based workflows and will benefit a wide spectrum of readers of the journal. If the authors could make the following two changes, it will further improve this review:

1.     Please include the label-free single cell analysis using DDA and DIA (such as timsTOF)-based technologies. 

2.     Please provide a table summarizing the technologies reviewed in the manuscript.    

Reviewer 2 Report

As the title already states, this review article covers "Advances in Mass Spectrometry-based Single Cell Analysis". In this context, several techniques that can be applied for the characterization of cellular content (proteins or metabolites) at the single-cell level are covered: TMT-based proteomics, mass cytometry, secondary ion MS, MALDI MS, and single-cell metabolomics approaches. The review starts with a brief discussion on why single-cell information is important and also briefly covers non-MS-based methods.

In general, I found the manuscript an interesting read. The choice of topics is relevant, the review summarizes different areas where mass spectrometry makes an impact on single-cell analysis. With few exceptions highlighted below, the text is clearly written, and the English is at a very high level.

My main criticism is that the review is mostly descriptive in nature and provides little context and perspective on the individual methods. Each chapter starts with a nice overview of the general methodology and then moves to single-cell applications. However, since this is a biology journal, the impact of each technique on generating new biological findings should be emphasized more. Expanding in this direction would make the article more impactful.

I will pick the section on TMT-based proteomics as an example, because this is the area I am most familiar with. With respect to single-cell applications, the authors discuss work from the Slavov and Kelly groups and cite some references related to methods and applications. However, to the non-expert reader who is not yet familiar with these papers it remains unclear what proteome coverage is achievable with these methods at present and how this compares to traditional bulk proteomics. For example, it is somewhat routine to cover five to ten thousand proteins from larger amounts of input material, while the numbers for single-cell applications are in the range of 1000 proteins. What kind of biological conclusions can be drawn from this proteome coverage? On a more technical side, it should also be pointed out that the nanoPOTS method uses custom-made equipment and is therefore not easily transferable to other labs.

Along the same lines, other sections should be expanded so that the reader can better understand which applications are proof-of-principle experiments or prototype instrument designs and which are already commercialized or more likely to be easily accessible in the future. For CyTOF, the availability of antibodies and their validation remains an issue. For SIMS and MALDI-MS, spatial resolution comes at the cost of reduced sensitivity, but what is the actual impact on proteome-coverage for "real-life" applications, and what new biology has been uncovered using single-cell approaches as opposed to bulk methods?

Finally, in the Conclusion section the topic of sample preparation/processing comes up repeatedly, but is underrepresented in the rest of the article. This would be another viewpoint to add to extended discussions.

Additional comments:

Page 4, line 153: The term "small mass ions" is imprecise. Please explain.

Page 6, line 231: "a laser beam with an energy-absorbing matrix ..." > Better use "a laser beam and an energy-absorbing matrix"

Page 6, line 238: A "QTOF" is not a mass analyzer, it is a type of instrument or instrument design combining quadrupole and TOF analyzers.

Page 6, line 243-244: The meaning of the sentence "MALDI MS is especially useful to analyze a mixture of analytes ..." is not clear, especially because LC-MS methods are discussed in other context in the article.

Page 7, line 285: wildly > widely

Generally, chapter titles are somewhat inconsistent: "TMT-Assisted Single-Cell Proteomics" is quite informative and descriptive, while "SIMS" or "MALDI" are not. These titles should be expanded.

Reviewer 3 Report

This review summarizes recent advances in MS-based single-cell analysis methods and their applications in biology and medicine. I think this work will be interesting to a reasonable number of readers working in the field of Mass spectra and single cell analysis. Anyway, a major revision is required and the following questions should be addressed.

1, Please investigate the single cell analysis by use of electrospray ion mass spectrometry, imaging MS,  laser-based mass spectrometry and inductively coupled plasma mass spectrometry.

2, A major, longstanding goal of scientists is to map protein expression, gene analysis, metabolic analysis in single cells.   Please investigate and add more text on the applications, for example metabolomics and proteomics analysis.

3, A number of different approaches have been developed to sample single cells, please investigate and discuss the cell sampling, throughput, and coverage.

4, Generally, a review contains 4-8 figures, and please add 3-5 figures in this manuscript.

5, Please cite the following reviews, 

Mass spectrometry-based strategies for single-cell metabolomics, Mass Spec Rev. 2023; 42: 67–94.

Single-cell mass spectrometry, Trends in Biotechnology, 2022, 40,1374-1392

Recent advances in single-cell analysis by mass spectrometry, Analyst, 2019,144, 824-845

Round 2

Reviewer 2 Report

The authors have done a great job in preparing this substantially revised version. In my opinion, they have addressed all the comments of the reviewers, and the additional content adds value to the review.

Minor comments: In the new parts of text, some typos should be corrected, e.g. at the proof stage. Examples are given below, together with some other suggestions.

Line 114: "DDA methods has" > method has or methods have

Line 169: Slavov's group or the Slavov group

Line 194: The abbreviation nanoPOTS is spelled out here, although the term is now already introduced earlier (line 96).

Line 310: nanopary > nanospray

Line 388: "falls their m/z values" > puts their m/z values

Line 449: body fluids, not bodily fluids

Line 449: Out of curiosity, I checked this reference, I don't see any data related to fish in #190. This reference is also cited in the same sentence for biofilm charaterization, which seems more appropriate. Please check.

Line 485: background (singular)

Line 658: a subpopulation or subpopulations

Line 689: workflows
